# There Is Hope in Safety Promotion! How Can Resources and Demands Impact Workers' Safety Participation?

**Simona Margheritti** [1,*] **, Alessia Negrini** [2] **and Sílvia Agostinho da Silva** [3]

1    Department of Psychology, University of Milano-Bicocca, Building U6, 20126 Milano, Italy
2    IRSST-Institut de Recherche Robert-Sauvé en Santé et en Sécurité du Travail, 505 Blvd. De Maisonneuve Ouest, Montreal, QC H3A 3C2, Canada; alessia.negrini@irsst.qc.ca
3    Department of Human Resources and Organizational Behavior, Iscte-Instituto Universitário de Lisboa, Av. das Forças Armadas, 1649-026 Lisbon, Portugal; silvia.silva@iscte-iul.pt
*    Correspondence: simona.margheritti@unimib.it

**Abstract:** Promoting workplace safety is crucial in occupational health and safety (OHS). However, existing studies have primarily concentrated on accident prevention, overlooking the role of resources in encouraging safety. This research investigates the impact of a personal resource, namely hope, on safety participation, considering its interaction with job resources and job demands using the Job Demands-Resources (JD-R) model in the context of safety. A cross-sectional study was conducted in a large company managing European shopping centers ($N$ = 262). Of the sample, 52.3% of participants were female. Data were collected through an online questionnaire and analyzed using model 92 of Andrew F. Hayes' Process Macro to test the hypothesized moderate serial mediation model. Our results highlighted that (1) hope directly correlates with safety participation, (2) hope and job dedication mediate the relationship between autonomy and safety participation, and (3) high job demands can undermine the beneficial effects of resources (i.e., autonomy, hope, and job dedication) on safety participation. These results suggest that workers with personal resources like hope are more likely to engage in safety practices, displaying increased dedication and focus on safety. However, excessive job demands can challenge the effectiveness of these resources in promoting safety participation. This study offers a novel perspective by integrating safety participation into the JD-R model framework.

**Keywords:** safety participation; hope; job resources; job dedication; JD-R model

## 1. Introduction

Promoting a safer work environment is one of the most significant challenges for organizations. Despite several interventions, the number of occupational accidents and injuries remains high [1]. Several studies deepened the topic of occupational safety and health (OHS), trying to solve the problem [2–7]. However, most of the studies in this area have only focused on safety prevention, investigating the factors leading workers to incur accidents and injuries or not.

While the findings of these studies are crucial, some scholars argue that the absence of adverse safety outcomes, such as accidents and injuries, does not necessarily indicate the presence of safety within organizations [2,8]. Accidents and injuries should be evaluated to determine the absence of workplace safety [2]. In fact, accidents or injuries do not occur in all circumstances, even when workers do not behave appropriately. Accidents typically depend on various elements, such as dangerous behaviors and underlying organizational deficiencies, that frequently coexist but do not always result in an accident [8]. Moreover, particularly concerning micro-accidents, workers often fail to report them, leading to their exclusion from record-keeping systems. This, in turn, results in an underestimation of the issue, limiting the analysis to the visible aspect rather than addressing the broader problem.

It is, therefore, necessary to broaden the perspective and focus on more informative behaviors that suggest both the existence and absence of safety [2,9], such as safety participation indicators. As Beus and colleagues affirm [2] these behaviors represent better the structure of safety and can be used as proximal indicators that diagnose a lack of safety before an actual injury is begun. Safety participation refers to proactive and voluntary behaviors that are not mandated by the worker's role and have an equal impact on workplace safety (e.g., attending safety meetings or assisting colleagues in hazardous conditions) [9]. Since they are non-mandatory safety-related behaviors, they are more influenced by the worker's motivation, knowledge, and resources to implement them. As a result, their enactment is not related to workers' shortcomings or mistakes but more to their resources. Following these premises, the present research investigates the role of a specific personal resource (i.e., hope) in promoting safety participation [7]. In addition, its interrelations with a specific job resource (i.e., autonomy), job dedication, and job demands were explored through the lens of the Job Demands-Resources (JD-R) [7,10–12] applied to safety.

## 2. Theoretical Background and Hypothesis Development

### 2.1. Hope: Definition and Antecedents

Snyder [13] (p.8) defined hope as a "positive motivational state that is based on an interactively derived sense of successful (a) agency (goal-directed energy) and (b) pathways (planning to meet goals)." As a psychological construct, hope is made up of three fundamental conceptual foundations: agency, paths, and objectives. The will to achieve the intended or desired consequence might be considered the agency component of hope [13–15]. As a result, hope entails the ability or motivation to achieve a goal. Furthermore, hope includes paths consisting of identifying goals and subgoals and different routes to reach these goals. Hopeful workers use contingency planning to anticipate hurdles to reaching goals or subgoals and develop numerous paths to achieving the desired outcome [13]. In other words, hope can be considered the will to succeed and the ability to recognize, clarify, and pursue a path to success [13]. Previous studies supported the association between hope and positive organizational outcomes, such as success, financial performance, employee retention, and job satisfaction [16], desirable work attitudes, such as job satisfaction and organizational commitment [17], and work happiness [18]. In this sense, the idea that being able to set tangible goals for oneself and achieve them helps the worker feel well and be successful has been confirmed. Despite these promising results, only a few studies were conducted to investigate hope's role in promoting safety.

Only two studies carried out with different samples of workers by Bergheim et al., [19,20] showed that hope was significantly related to safety climate. Specifically, air traffic controllers showed that the ability to redirect efforts to reach their objectives (i.e., hope) impacts the safety climate. Recently, the same significant results about hope, were found in a study from Saleem and colleagues [21] among 345 Malaysian construction workers, showing that hope can significantly and positively affect both safety performance indicators. Other noteworthy results are summarized in Margheritti et al. [22] literature review, indicating that hope, as a sub-dimension of PsyCap, is directly and indirectly associated with safety performance. Specifically, two studies included in the review and carried out in China showed that hope was highly and positively associated with safety participation in a sample of 400 frontline workers in coal mines [23] and among construction workers [24]. Instead, He et al. [25] observed that hope was not directly related to safety behaviors (i.e., compliance and participation) but indirectly to safety participation by communication competencies. However, only three of the twenty studies included in Margheritti et al.'s [22] review investigated the specific role of hope, presenting several limitations.

Considering that hope is associated with several positive organizational outcomes (e.g., financial performance, employee retention, and job satisfaction [16], job satisfaction and organizational commitment [17], and work happiness [18]) by helping workers recognize and organize their goals and achieve them effectively, it was hypothesized that it could also improve safety participation, fostering people to be more determined to participate in

safety. From our perspective, workers must comply with safety rules and protocols because they are mandatory (i.e., safety compliance), but they can choose to participate in safety without any constraint. Thus, hope could effectively improve the voluntary nature of safety participation and the motivational desire of workers to act safely.

**H1:** *Hope is positively associated with safety participation.*

*2.2. The Motivational Process of the Job-Demands Resources Model Applied to Safety: The Role of Hope*

The JD-R model [10–12,26] is one of the more effective models in the field of organizational psychology because it can predict positive and negative outcomes linked to workers' health and performance. According to the model, job demands and resources give rise to two independent processes, health impairment, and motivational processes, that predict organizational outcomes. Excessive job demands, in the long run, lead to constant arousal, which can lead to exhaustion, psychosomatic symptoms, and health damage (health impairment process). The presence of job resources, such as autonomy, contributes to workers' motivation (motivation process).

The added value of this model in the OHS field is that it integrates job resources and job demands. For instance, the interactive effect between job demands and job resources leads to an attenuation of the negative effect of job demands on stress [10,27]. An additive effect has also been shown by Bakker et al. [28]. Indeed, demanding working conditions in the presence of job resources translate into work motivation and commitment within the organization [26]. The latter scenario represents an example of an occupational health situation in which workers can perform well, learn, and develop their skills [10,12].

Evidence in the literature shows that the JD-R model can also be applicable and valuable within the safety context [29,30]. Beus et al. [2], in their integrated safety model (ISM), summarized several results that support the relationship between job resources and safety outcomes. Some of them are from Nahrgang and colleagues [7], who, in a meta-analysis of 203 independent samples, showed the presence of relations between job demands and resources and workplace burnout, engagement, and some adverse safety outcomes such as accidents, injuries, adverse events, and unsafe behaviors. However, it is still unclear how the role of personal resources in the JD-R model applies to safety. Although recent developments of the JD-R model also recognized the idea that human behaviors result from the interaction between personal characteristics (such as personal resources) and environmental factors (such as job demands) [11,31,32], few previous studies have deepened this issue.

Referring to the motivational process of the JD-R model, we hypothesized that having high-quality job resources (such as autonomy) helps workers to be hopeful (i.e., express persistent pursuit of goals and the proactive identification of pathways), be more dedicated at work, and, in turn, behave more safely. From the theoretical point of view, this idea is also supported by the conservation of resources theory (COR) [33,34], which assumes that resources tend to accumulate. Thus, workers working in a resourceful environment are likely to develop feelings of hope [35] about their future at work. In turn, this personal resource will be positively related to their job dedication. As part of work engagement, job dedication is defined as being intensely involved in one's work and feeling a sense of significance, enthusiasm, inspiration, pride, and challenge [36]. From this perspective, it could be strictly related to hope. Indeed, hope, expressed by the constant pursuit of goals and the proactive identification of pathways, may support the dedication to reach goals [37].

Thus, we hypothesized that:

**H2:** *Autonomy promotes safety participation by increasing workers' hope and job dedication.*

### 2.3. The Job-Demands Resources Model Applied to Safety: The Role of Job Demands

One of the key dimensions of the JD-R model are the job demands. As job demands increase to the point of workload, damaging effects on psychological and physical health, performance, and effort should be reflected [38]. Scholars have shown that excessive job demands stress workers and contribute to burnout, negatively impacting their work engagement [39]. A high workload could also increase the likelihood of workers enacting risky safety behaviors [40], have a detrimental effect on safety performance [41], and increase fatigue associated with the increased risk of incidents [30,42]. All these results indicate that workload has a direct and adverse effect on safety performance; thus, the higher the job demands, the lower the likelihood that workers will perform safely. At the same time, other studies on this field point out that job demands could also be challenging and opportunities to learn, achieve, and show competence, leading workers to behave better and viewing demands as opportunities for mastery, personal advancement, or future rewards [11].

Thus, in this research, we aim to delve deeper into this topic by focusing on the role of job demands in the relationship between resources and safety participation. Specifically, this investigation seeks to understand whether job demands moderate the indirect relationship between resources and safety participation, mediated by job dedication. We hypothesize that when the workload is excessively high, workers may allocate all their energy to coping, leaving insufficient resources for safe behavior. This idea is supported by Hobfoll's [33,34] conservation of resources theory (COR). According to Hobfoll [33,34], persistent exposure to high job demands could amplify their impact on adverse organizational outcomes. The accumulation of job demands increases the likelihood that all available energy resources on a given day will be depleted, leaving little to devote to other types of behavior.

In our case, a heavy workload is expected to weaken the positive association between resources and safety participation, rendering the presence of job dedication insufficient to translate into safety participation. Consequently, a lack of cognitive and energetic resources for safety concerns could lead to suboptimal safety performance. We hypothesize that elevated workload levels may attenuate the strength of the relationship between resources and safety participation.

**H3:** *High job demands reduce the strength of the relationship between autonomy, hope, job dedication and safety participation.*

To sum up, the hypothesized conceptual model is shown in Figure 1.

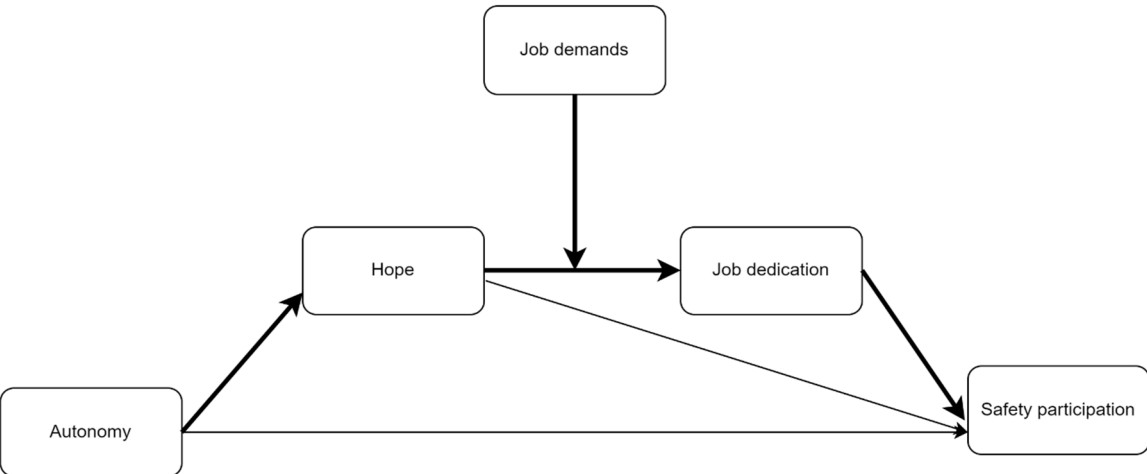

**Figure 1.** Conceptual model. Note. The relationship shown in bold refers to the hypothesis H3.

## 3. Method

### 3.1. Recruitment Process

Participants were recruited from a company managing European shopping centers. No specific inclusion or exclusion criteria were set, except for having an employment contract with the company. First, the researchers presented the study to the employer and to the human resources department, who, in turn, spoke to the workers. They were then provided a web link containing highlights of the study and access to the online survey. Participants had the voluntary choice to participate or not and decided the timing for completing the survey. All data were collected anonymously, and the survey took 15 min on average. Following Leedy and Ormrod's [43] recommendations, validated with an online sampling adequacy verification service, the recruitment process provided a representative sample of all the company's staff (N $\simeq$ 650).

### 3.2. Measures

Autonomy (4 items, $\alpha$ = 0.80) was measured by the Work Design Questionnaire (WDQ); [44]. This is a Likert scale instrument on a five-point ordinal scale from 1 "strongly disagree" to 5 "strongly agree". Example of item: "My job provides me with significant autonomy in making decisions".

Hope (3 items, $\alpha$ = 0.72) was measured using the relative dimension of the Psychological Capital Questionnaire (PCQ-24) developed by [45]. This is a Likert scale instrument on a five-point ordinal scale from 1 "strongly disagree" to 5 "strongly agree". Sample items include "I can think of many ways to reach my current work goals".

Job dedication (3 items, $\alpha$ = 0.87) was assessed through the relative dimension of the short version of the Utrecht Work Engagement Scale (UWES-9) [46]. Example of item: "I am proud of the work that I do." The response scale ranged from 1 "never" to 5 "always".

The Job demands (4 items, $\alpha$ = 0.85) variable from the Management Standards Health and Safety Executive (HSE) indicator tool [47] was used to measure work-related issues, such as workload. Respondents answered eight items on a scale ranging from 1 "never or strongly disagree" to 5 "often or strongly agree", with lower scores indicating higher job demands.

Safety participation (4 items, $\alpha$ = 0.81) was assessed using the safety performance scale developed by Griffin and Neal [9] This is a Likert scale instrument on a five-point ordinal scale from 1 "strongly disagree" to 5 "strongly agree". Examples of items: "I promote the safety program within the organizations" or "I put an extra effort to improve the safety of the workplace".

### 3.3. Data Analysis

The data were processed using SPSS Software 28.0 (IBM Corp., 2021, Armonk, NY, USA). Initially, descriptive statistics were calculated to know the characteristics of the sample better. Secondly, the scales were calculated through the average item value calculation, which is the sum of all items on the same scale divided by the total number of items. Then, correlation analyses were conducted to explore the associations between the studied variables. ANOVAs were executed to test whether there were differences in our DV (safety participation) based on the socio-demographic characteristics of the sample. Since no significant difference was found in our dependent variable (DV) based on sample characteristics, no covariates were included in our subsequent analyses.

Following the PROCESS macro developed by Peacher and Hayes [48], model 92 of Andrew F. Hayes' Process Macro (V. 4) was used to test the hypotheses. Model 92 allows the testing of a moderated serial mediation model. In this specific model, job demands are allowed to moderate the direct path from hope (M1) to job dedication (M2), which, in turn, are the serial mediators in the relationship between autonomy (independent variable; IV) and safety participation (DV).

## 4. Results

### 4.1. Sample

The sample was composed of 262 workers (52.3% female) from a company that manages shopping centers. Of these, 4.6% were under 29 years old, 46.9% were aged 30–39, 35.5% were aged 40–49, and the remaining were over 50 years old. The majority (88.5%) had completed at least a university degree. The employment contract was full-time for 96.6% of the workers. The majority of workers (62.9%) were based in central offices, while the others worked in shopping center offices.

### 4.2. Preliminary Analyses

Table 1 shows means, standard deviations, and correlations for studied variables. The correlation matrix shows that DV investigated (or safety participation) is not always significantly associated with IV. Nevertheless, according to Peacher and Hayes [48], there may be an indirect relationship between the variables through the action of an intervening mediator. Therefore, the PROCESS macro, precisely the Bootstrap method, will be used to test the mediation hypotheses.

**Table 1.** Correlation matrix of the studied variables.

| | Variable | M | SD | 1 | 2 | 3 | 4 |
|---|---|---|---|---|---|---|---|
| 1. | Autonomy | 3.49 | 0.75 | | | | |
| 2. | Hope | 3.67 | 0.64 | 0.41 *** | | | |
| 3. | Job dedication | 3.78 | 0.74 | 0.39 *** | 0.55 *** | | |
| 4. | Job demands | 3.27 | 0.82 | −0.38 *** | −0.26 *** | −0.21 *** | |
| 5. | Safety participation | 3.75 | 0.63 | 0.04 | 0.26 *** | 0.26 *** | −0.17 |

Notes. *** $p < 0.001$, ** $p < 0.01$, * $p < 0.05$.

### 4.3. Model Testing

Model 92 of the SPSS Process Macro developed by Hayes [49] was used to test the three hypotheses. The effect of hope on safety participation (H1) was verified ($\beta = 0.18$, $se = 0.08$, $p = 0.02$). Additionally, we found support for the indirect effect of autonomy on safety participation through hope and job dedication (H2: $\beta = 0.05$, $se = 0.02$, CI 95%: [0.01;0.09]). The main path coefficients ($\beta$) of these models are shown in Figure 2. For the moderation effect, where job demands moderated the mediated relationship between resources and safety participation (H3), we found that under low workload (−1 SD), the serial mediation is significant, as well as when the workload is moderated (mean levels). However, under a high workload (+1 SD), the serial mediation became not significant (see Table 2). These findings supported moderation (H3) in the serial mediation relationships between variables.

**Table 2.** Indirect conditional effect of autonomy, hope, and job dedication on safety participation at the workload levels.

| | Indirect Effect on Safety Participation | | Bootstrapping | |
|---|---|---|---|---|
| | Point Estimate | BootSE | Lower 95% CI | Upper 95% CI |
| IV Autonomy | | | | |
| Low workload (−1 SD) | 0.0347 | 0.0197 | 0.0069 | 0.0839 |
| Medium workload | 0.0199 | 0.0104 | 0.0040 | 0.0447 |
| High workload (+1 SD) | 0.0025 | 0.0122 | −0.0180 | 0.0309 |

Note: IV = Independent variable.

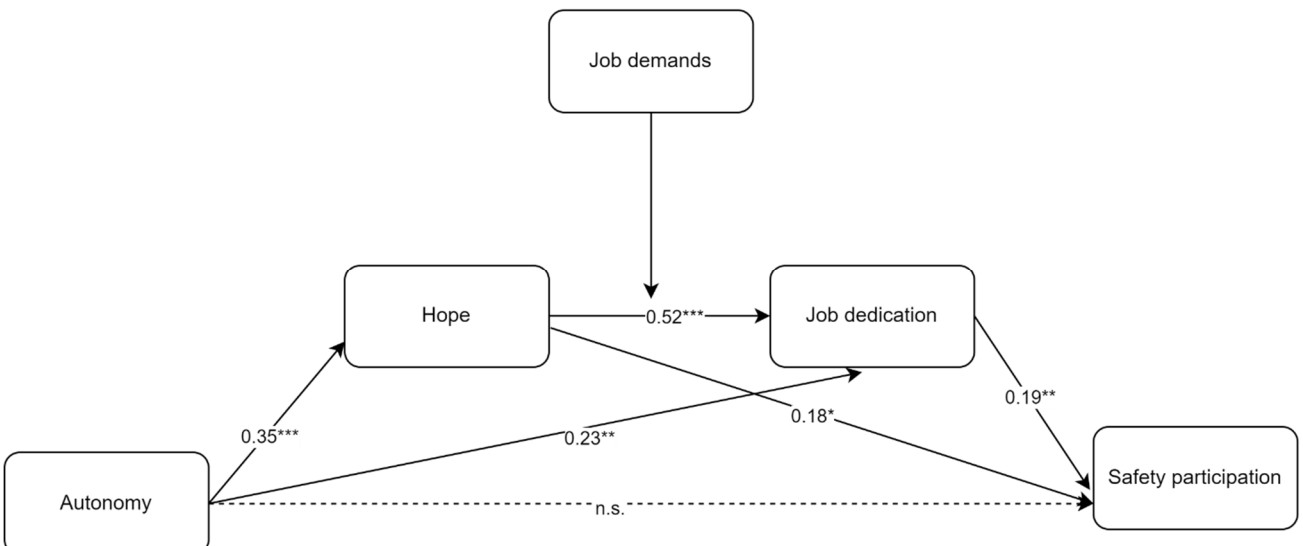

**Figure 2.** The main effect on safety participation with autonomy as the independent variable. Notes. *** $p < 0.001$, ** $p < 0.01$, * $p < 0.05$. Dashed lines indicated no significant relationship patterns.

## 5. Discussion

Within the framework of the JD-R model applied to safety [7], this study aims to assess the role of a specific personal resource (i.e., hope) in promoting workers' safety participation. The present study also investigates the interaction between hope, job resources, job dedication, and job demands in promoting it.

The most important result of this investigation is that hope directly promotes safety participation. In fact, the study of the role of personal resources in promoting safe behavior is a new topic, still little explored in the safety field [22]. Following Ye, et al.'s [23] study, we found a positive association between hope and safety participation. This finding indicates that being able to think about achievable objectives, the ways to achieve them, and possibly be able to experiment with alternative courses of action leads the employee to devote himself with greater probability to safety participation behaviors. Confidence in one's abilities is also transferred within the safety framework, encouraging the worker to take action to create a work environment that is as safe as possible, going beyond the simple respect of the rules linked to one's role (safety compliance).

In addition, our results showed that hope and job dedication (H2) together could explain the association between autonomy (as job resources) and safety participation. Specifically, having the possibility to manage tasks and timing at work makes workers hopeful, thus being able to identify goals and subgoals and different routes to those goals. More hopeful workers will also feel more dedicated to work. Having directed goals and planning to meet goals helps workers provide the willingness (dedication) to reach goals. Considering that job dedication, as part of work engagement, has been shown to be strongly associated with several positive organizational outcomes [11] the development of hope should be considered an important point for organizations that want to promote health and safety. Finally, hopeful and dedicated workers engage more frequently in safety participation behaviors. The findings align with the JD-R model [10,11] and COR theory [34,50], indicating the close and mutual relationship between job and personal resources. At the same time, these results broaden research on the antecedents of personal resources [51]. The fact that the direct impact of autonomy on safety performance is not significant means that it is not enough to perceive to have the time and opportunity to enact voluntary behaviors. Instead, it is also necessary to be accompanied by the perception of being able to organize and achieve goals and being dedicated to work.

Lastly, although job (i.e., autonomy), personal (i.e., hope), and motivational (i.e., job dedication) resources seem to play a significant role in promoting safety participation,

when the workload is too high, the virtuous effect of these resources disappears (H3). The impact of job demands enforces the role of organizational context, indicating that it is impossible to isolate the role of personal resources from other job characteristics. As COR theory [34,50] postulates, constantly being subjected to excessive job demands may negatively affect organizational outcomes. In fact, accumulating job demands raises the chance that all of one's energy reserves would be depleted on any given day, leaving little energy for other kinds of behavior. In our study, high job demands weaken the favorable link between job dedication and safety participation, rendering resources insufficient to achieve safety participation. Therefore, poorer safety performance results from a lack of cognitive and physical resources to dedicate to safety-related concerns.

## 6. Limitations, Further Research, and Implications

The results stemming from this investigation must be interpreted considering certain limitations that raise issues for future studies. Firstly, the use of self-reported data increases the likelihood of social desirability and common method bias [52]. Future research should investigate the same phenomenon by including an analysis of objective dates related to safety behaviors (e.g., real participative behaviors from the organizational report) that eliminate this issue, strengthening our results. The study's cross-sectional design is a second limitation, which does not allow for testing causality between variables. Future longitudinal studies are needed to verify the model. To better understand the role of personal resources (such as hope) within the JD-R Model applied to safety, future research should also explore its association with other job resources (e.g., safety climate, support from colleagues or supervisors), work attitudes (e.g., general work engagement or job satisfaction), and objective safety outcomes (e.g., accident rates, number of injuries, and fatalities). The model tested in this study should be replicated using a sample of blue-collar workers, generally characterized by lower levels of education, less autonomy, and the performance of manual tasks. Replication of the study with this demographic sample would allow us to determine whether our results are valid and generalizable in different working populations. Furthermore, it would be of future research interest to explore the significance of hope within the psychological capital framework [53,54], particularly in its interplay with the other three components, namely self-efficacy, optimism, and resilience. Such an inquiry would not only elucidate the distinct contributions of hope but also shed light on its interrelationships with the other elements in facilitating safety-related behaviors.

Despite the above limitations, the present study makes significant theoretical and practical contributions. Firstly, it is one of the few studies, along with Nahrgang et al. [7], to analyze the issue of promoting safety behaviors within the theoretical framework of the JD-R model [7]. The present research highlights how job and personal resources interact within its motivational process, giving rise to better safety performance. This suggestion means the JD-R model can also explain and predict these positive safety outcomes. Secondly, through this study, we can confirm the positive role of workers' hope in implementing safety participation behaviors. In this sense, it becomes possible not only to focus on the variables that lead workers to incur occupational accidents and injuries but also on the personal resources that, if developed and implemented, can give rise to virtuous safety behaviors. The practical implication consistent with these results is the need for safety training programs to focus not only on safety procedures or rules but also on improving workers' positive attitudes (e.g., hope). As part of acquiescence requirements, safety training is often mandatory. However, when safety training is only negatively oriented, compliance-based, or implemented due to accidents and injuries [55], it can have a limited influence on worker motivation. As a result, workers will rarely behave safely in a proactive, agentic, and intentional way. Human resource management and safety professionals could first promote autonomy at work and then implement practices or training to develop hope into routine safety training events [13,16,18,53,54], increasing their effectiveness greatly. In addition, organizations should address and effectively manage job demands that have the potential to undermine all the efforts made to promote safety.

## 7. Conclusions

This study has enabled a deeper understanding of hope's key role in promoting safety participation within organizations and its relationship with autonomy, job dedication, and job demands. The findings of the study suggest that the ability to envision achievable goals, devise plans to attain them, and consider alternative actions—collectively referred to as 'hope'—increases the likelihood of workers engaging in safety behaviors that go beyond mere compliance with safety regulations, known as 'safety participation'. Furthermore, safety participation is fostered by employee hopefulness, which is linked to job dedication and empowered by the ability to make decisions about one's work, utilizing one's skills and competencies—referred to as 'autonomy'. However, while job autonomy, personal hope, and motivational factors like job dedication significantly enhance safety participation, an excessive workload can undermine the positive influence of these resources. Prolonged exposure to high job demands can deplete energy reserves, leaving inadequate resources for safety participation, thereby highlighting the necessity of managing workloads to optimize safety outcomes. Therefore, it is crucial not to dissociate the evaluation of job demands from the resources required to meet them. This integrated model can inform OHS interventions that aim to promote safe behaviors by achieving a balance between job demands—such as by reducing workload and enhancing autonomy—and considering workers' personal and motivational resources.

This study represents an innovative contribution to the literature on safety participation, investigating its promotion within the motivational process of the JD-R model. Our results suggest new opportunities for groundbreaking research focused on occupational health and safety (OHS) promotion through the lens of positive resources.

**Author Contributions:** Conceptualization, S.M. and S.A.d.S.; Methodology, S.M. and S.A.d.S.; Software, S.M.; Validation, S.A.d.S. and A.N.; Formal Analysis, S.M.; Investigation, S.M. and S.A.d.S.; Data Curation, S.A.d.S.; Writing—Original Draft Preparation, S.M.; Writing—Review and Editing, A.N. and S.A.d.S.; Visualization, S.M.; Supervision, S.A.d.S. and A.N.; Project Administration, S.A.d.S. All authors have read and agreed to the published version of the manuscript.

**Funding:** This research received no external funding.

**Institutional Review Board Statement:** The study was conducted in accordance with the Declaration of Helsinki and approved by the Research Evaluation Committee, Department of Psychology, University of Milano-Bicocca (RM-2020-352). The project was approved on 7 January 2021.

**Informed Consent Statement:** Informed consent was obtained from all subjects involved in the study.

**Data Availability Statement:** The data presented in this study are available on request from the corresponding author.

**Conflicts of Interest:** The authors declare no conflict of interest.

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
