# Peer review of "There Is Hope in Safety Promotion! How Can Resources and Demands Impact Workers’ Safety Participation?"

_safety, 2023_

Round 1

Reviewer 1 Report

Comments and Suggestions for Authors

This study offers a novel perspective by integrating safety participation into the JD-R model framework, but this article should correct the following minor errors to accept.

1. The abbreviation of "Job Demands-Resources Model" change to "JD-R Model" consistantly (P1, L15; P2, L57; P3, L108) and list in "Keywords".

2. There are several writing errors in the cited references at L108-109, L119-120, L126, L128, L166, L172, L173, L199, L207, L215, L227, L232, L262-263. Please correct them according to the format.

3. What is the meaning of IV (L232)? Chage "IV" to job demands based on Table 1.

4. The cited references no.3 and no.12 are wrong, please correct them as followed.

(Beus, J. M., McCord, M. A., & Zohar, D. Workplace safety: A Review and Research Synthesis. Organizational psychology review 20166, 352-381, doi:10.1177/2041386615626243.

Snyder, C.R. Handbook of Hope: Theory, Measures, and Applications; Academic Press, 2000; ISBN 9780080533063.)

Author Response

Dear Referee#1,

We would like to thank you for having accepted to review our article, and for your valuable comments and suggestions which allowed us to improve its quality. Below you will find your original comments followed by our responses, including the report of where in the manuscript we edited the text (page numbers and lines). All changes, suggested by you and referees #2 and #3 are highlighted in yellow.

We hope that the revised version of the manuscript is satisfactory for publication in the Safety journal.

With kind regards,

The authors

COMMENTS

ANSWERS

REV 1

This study offers a novel perspective by integrating safety participation into the JD-R model framework, but this article should correct the following minor errors to accept.

Thank you for your comments; all minor typos have been corrected following the directions.

1. The abbreviation of “Job Demands-Resources Model” change to “JD-R Model” consistantly (P1, L15; P2, L57; P3, L108) and list in “Keywords”.

The abbreviation has been corrected throughout the text and made consistent.

2. There are several writing errors in the cited references at L108-109, L119-120, L126, L128, L166, L172, L173, L199, L207, L215, L227, L232, L262-263. Please correct them according to the format.

All inaccuracies and errors related to the bibliography have been corrected.

3. What is the meaning of IV (L232)? Chage “IV” to job demands based on Table 1.

Suggested corrections have been implemented.

4. The cited references no.3 and no.12 are wrong, please correct them as followed.

(Beus, J. M., McCord, M. A., & Zohar, D. Workplace safety: A Review and Research Synthesis. Organizational psychology review 20166, 352-381, doi:10.1177/2041386615626243.

Snyder, C.R. Handbook of Hope: Theory, Measures, and Applications; Academic Press, 2000; ISBN 9780080533063.)

Thank you very much for your comments; the references no.3 and no.12 have been updated following the suggestions.

Reviewer 2 Report

Comments and Suggestions for Authors

Dear Authors,

I am grateful for the opportunity to review your scientific article. However, I have some specific concerns and suggestions to improve the clarity and comprehensibility of your work:

The abstract lacks essential information, such as a description of the study participants and details about the data processing methods. It is crucial to include these elements to provide readers with a clear understanding of the study's foundation.

While the Job Demands-Resources model is an exciting approach with a wide range of perspectives, it would be beneficial to translate the results into more traditional terminology. For instance, integrating concepts such as satisfaction, resilience, stress, and workload could provide a broader context for readers to relate to.

The conclusion seems accurate and applicable; however, your specific sample is quite specialized, as it was conducted with highly skilled subjects with high autonomy in jobs.

The methodology and data processing techniques should be detailed more extensively. The reference to "Model 92" appears insufficient for readers to understand your approach fully. Additionally, the calculation methods for the scales used in the study should be explicitly explained.

In conclusion, your article presents interesting findings and concepts but requires several improvements to enhance its clarity and accessibility to a broader audience. I recommend addressing the abovementioned points to make your work more comprehensive and reader-friendly.

Thank you for considering my feedback, and I look forward to seeing the revised version of your paper.

Comments on the Quality of English Language

All abbreviations, such as IV (Independent Variable) and DV (Dependent Variable), should be expanded upon when first introduced to ensure readers are not confused.

The article's clarity is hampered by issues with language usage, making it challenging for readers to comprehend your work. Proofreading and language editing may be necessary to improve the overall quality of the paper.

Author Response

Dear Referee#2,

We would like to thank you for having accepted to review our article, and for your valuable comments and suggestions which allowed us to improve its quality. Below you will find your original comments followed by our responses, including the report of where in the manuscript we edited the text (page numbers and lines). All changes, suggested by you and referees #1 and #3 are highlighted in yellow.

We hope that the revised version of the manuscript is satisfactory for publication in the Safety journal.

With kind regards,

The authors

COMMENTS

ANSWERS

REV 2.

  1. I am grateful for the opportunity to review your scientific article. However, I have some specific concerns and suggestions to improve the clarity and comprehensibility of your work:

We thank you for your comment, for taking the time to review our article, and for your valuable comments.

  1. The abstract lacks essential information, such as a description of the study participants and details about the data processing methods. It is crucial to include these elements to provide readers with a clear understanding of the study’s foundation.

We thank the reviewer for the comment; the requested information has been included in the abstract to provide greater clarity (p.1).

  1. While the Job Demands-Resources model is an exciting approach with a wide range of perspectives, it would be beneficial to translate the results into more traditional terminology. For instance, integrating concepts such as satisfaction, resilience, stress, and workload could provide a broader context for readers to relate to.

All the concepts recommended by the reviewer, including satisfaction, resilience, stress, and workload, have long been integrated into the JD-R (Job Demands-Resources) model. In our view, the JD-R model stands out as the most effective theoretical framework for this very reason. This model effectively synthesizes various theories to elucidate the correlations between resources (e.g., resilience) and demands (e.g., workload), and their direct or indirect relationships with workers’ health and organizational outcomes, such as work stress and job satisfaction. From this perspective, we conducted our analyses, examining how resources (autonomy and hope) relate directly and indirectly to safety performance, which serves as a positive organizational outcome.

We recognized that the concepts suggested by referee#2 are intriguing and aligned with our study. However, we would prefer not to integrate further concepts in the results because they were not explicitly addressed in this specific research endeavor.

  1. The conclusion seems accurate and applicable; however, your specific sample is quite specialized, as it was conducted with highly skilled subjects with high autonomy in jobs.

We thank the reviewer for the comment and agree with what he/she wrote. In fact, we are replicating the same model by conducting a study involving blue collar, i.e., low-education workers doing a manual labor with low levels of autonomy. This second study should show if the relationships between studied variables hold up, or if significant differences emerge. We added this point in the limitation section (p. 8, lines 329-333).

  1. The methodology and data processing techniques should be detailed more extensively. The reference to “Model 92” appears insufficient for readers to understand your approach fully. Additionally, the calculation methods for the scales used in the study should be explicitly explained.

The PROCESS macro is a modification to statistical programs like SPSS that computes regression analyses containing various combinations of mediators, moderators, and covariates. The model, 92 of Andrew F. Hayes’ Process Macro (V. 4), is a model implemented in SPSS to test moderated serial mediation analyses. A sentence was inserted at the end of the data analysis paragraph to explain better what Model 92 is for (p. 5, lines 231-235).

The scales were calculated through the average item value calculation, which is the sum of all items on the same scale divided by the total number of items. (line 223-224)

  1. In conclusion, your article presents interesting findings and concepts but requires several improvements to enhance its clarity and accessibility to a broader audience. I recommend addressing the abovementioned points to make your work more comprehensive and reader-friendly.

Thanks for the suggestions; the comments have been carefully followed and resolved to make the work more straightforward and understandable.

  1. All abbreviations, such as IV (Independent Variable) and DV (Dependent Variable), should be expanded upon when first introduced to ensure readers are not confused.

Thank you for the suggestions, we added specifications regarding all the abbreviations in the manuscript.

  1. The article’s clarity is hampered by issues with language usage, making it challenging for readers to comprehend your work. Proofreading and language editing may be necessary to improve the overall quality of the paper.

Thank you for your comments and suggestions to make the text smoother and more fluent. English has been improved using special software (i.e., Grammarly premium) and reading by a colleague expert in the OHS field and fluent in English.

Reviewer 3 Report

Comments and Suggestions for Authors

In the abstract, the sentences related to methods and results should be clearly elucidated. Please provide a clear and organized structure.

In the introduction, it’s a bit weird to have the two sentences in the first paragraph.

In the introduction, “It is therefore necessary to change perspective and focus on more informative…”, whether it is appropriate to use a strong wording here, “necessary to change”. Or you would like to foster/ enhance something.

In 3.1 Method, you should state the recruitment process and participant criteria instead of stating “The sample was composed of 262 employees (52.3% female) of company that manages shopping centres…”. This should be stated in the results section.

Regarding the sample size, how did you define it? What’s the representative of this population? All these should be clearly demonstrated.

The conclusion is too brief. Please improve it. 

Author Response

Dear Referee#3,

We would like to thank you for having accepted to review our article, and for your valuable comments and suggestions which allowed us to improve its quality. Below you will find your original comments followed by our responses, including the report of where in the manuscript we edited the text (page numbers and lines). All changes, suggested by you and referees #1 and #2 are highlighted in yellow.

We hope that the revised version of the manuscript is satisfactory for publication in the Safety journal.

With kind regards,

The authors

COMMENTS

ANSWERS

REV 3.

  1. In the abstract, the sentences related to methods and results should be clearly elucidated. Please provide a clear and organized structure.

Thank you for the valuable comment, the abstract has been improved adding some information regarding the methods. In addition, the sentence highlighting the 3 main results was revised.

  1. In the introduction, it’s a bit weird to have the two sentences in the first paragraph.

The two sentences in the first paragraph have been supplemented with the following sentences in the same paragraph (p1 lines 29-31).

  1. In the introduction, “It is therefore necessary to change perspective and focus on more informative…”, whether it is appropriate to use a strong wording here, “necessary to change”. Or you would like to foster/ enhance something.

We thank the reviewer for the valuable comments. Indeed, in this sentence we wanted to make explicit the need to broaden the perspective of safety studies and interventions by including interest in the role of positive resources. We have rephrased the sentence to highlight more the broadening of the perspective rather than its radical change. The sentence in question has been reworded as “It is therefore necessary to broaden the perspective” (page 2, line 46)

  1. In 3.1 Method, you should state the recruitment process and participant criteria instead of stating “The sample was composed of 262 employees (52.3% female) of company that manages shopping centres…”. This should be stated in the results section.

A specific section dedicated to the recruitment process was added in the Method section (paragraph 3.1, p. 5), while the description of the sample was moved to the results (paragraph 4.1., page 6).

  1. Regarding the sample size, how did you define it? What’s the representative of this population? All these should be clearly demonstrated.

The total number of employees in the company in question is approximately 650. In determining the sample size, two criteria were employed.

1) The first criterion was based on the recommendation put forth by Leedy and Ormrod (2015). They suggested that when the population size is in the vicinity of 500 (with a range of approximately 100), 50% of the population should be sampled.

Leedy, P. D., & Ormrod, J. E. (2015). Practical research. Planning and design (11th ed.). Boston, MA: Pearson. 

https://pce-fet.com/common/library/books/51/2590_%5BPaul_D._Leedy,_Jeanne_Ellis_Ormrod%5D_Practical_Res(b-ok.org).pdf

2) Additionally, we employed an online sampling adequacy verification service. By specifying a total population size of 650 employees, a 95% confidence interval, a 5% margin of error, and a 50% population proportion, the ideal sample size was determined to be about 242 individuals.

https://www.calculator.net/sample-size-calculator.html

In addition to this, all employees were invited to participate in the survey, partly to ensure the representation of the company's workforce as realistically as possible. Nevertheless, an analysis of the descriptive statistics demonstrates that our sample is a faithful representation of the population we sought to survey.

  1. The conclusion is too brief. Please improve it. 

The conclusion paragraph has been expanded upon, taking up the key findings of the study (p 9, lines 364-379)

Round 2

Reviewer 1 Report

Comments and Suggestions for Authors

I accept this revised version after improvement and error correction.

Author Response

We thank the reviewer for the precious comments and for considering our paper valuable and interesting

Reviewer 2 Report

Comments and Suggestions for Authors

Dear Authors,

Thanks for the improvement.

Bests

Author Response

(The authors gave the same response as above.)

Reviewer 3 Report

Comments and Suggestions for Authors

N/A

Comments on the Quality of English Language

English editing is required before published. 

Author Response

We thank the reviewer for the precious comments and for considering our paper valuable and interesting. After incorporating the suggested changes, the English text underwent further revision. This process was facilitated by error-detection tools such as Grammarly Premium and ChatGPT. Subsequently, the manuscript was reviewed by an expert in the field of OHS, who affirmed that it is now entirely comprehensible and stylistically correct.